# Short Communication: Low Prevalence of Clinically Important Antibiotic-Resistant Strains among Non-Pathogenic Genera of the Tribe *Klebsielleae*

**DOI:** 10.3390/foods11152270

**Published:** 2022-07-29

**Authors:** Arkadiusz Józef Zakrzewski, Wioleta Chajęcka-Wierzchowska, Anna Zadernowska

**Affiliations:** Department of Industrial and Food Microbiology, University of Warmia and Mazury, Plac Cieszyński 1, 10-726 Olsztyn, Poland; arkadiusz.zakrzewski@uwm.edu.pl (A.J.Z.); anna.zadernowska@uwm.edu.pl (A.Z.)

**Keywords:** *Hafnia* sp., *Serratia* sp., *Klebsielleae*, antibiotic resistance

## Abstract

*Hafnia* sp. and *Serratia* sp. belong to the Tribe *Klebsielleae*; although they are not considered pathogenic bacteria, there are many documented cases of diseases caused by these microorganisms. The aim of this study was to determine the antibiotic resistance profiles of strains belonging to the genus *Hafnia* and *Serratia* isolated from fish and shrimps. Phenotypic antibiotic resistance was determined using the semi-automatic Vitek 2 system (bioMérieux, Marcy-l’Étoile, France), while the presence of the extended-spectrum beta-lactamase, AmpC beta-lactamases, *Klebsiella pneumoniae* carbapenemases and Metallo-β-Lactamase producing strains were determined using the MIC Test Strip (Liofilchem, Roseto degli Abbruzzi, Italy). As a result of the conducted research, it was observed that a vast number of *Hafnia* sp. strains were resistant to cefalexin (84.61%), while *Serratia* sp. Strains to cefuroxime (79.41%) and nitrofurantoin (85.29%). In addition, it was observed that of all strains, only one had an ability to produce enzymes typical for β-lactamase-producing Enterobacterales. Although the strains of *Hafnia* sp. and *Serratia* sp. isolated from fish and shrimp are not characterized by frequent resistance to antibiotics, taking into account the constantly growing number of antibiotic-resistant strains, this may be a problem in the future, mainly due to gene transfer through mobile genetic elements and the acquisition of resistance expressed phenotypically through contact with stress factors. Therefore, studies monitoring the antibiotic resistance profile of these species should be carried out on a regular basis.

## 1. Introduction

The tribe *Klebsielleae* belongs to the order Enterobacterales, which includes four genera: *Klebsiella*, *Enterobacter*, *Hafnia* and *Serratia* [1,2]. Although species of the genera *Hafnia* and *Serratia* are not considered clinically significant, there are increasing reports of various diseases caused by these genera [3,4,5]. *Hafnia* and *Serratia* sp. are Gram-negative facultative anaerobic bacteria that grow over a wide range of temperatures and substrates, including plant surfaces, soil, water and food products such as fruits and fish [6,7]. *Hafnia* sp. occurs mainly in animal products, including minced and vacuum-packed beef and pork products, dairy products and fish. Vegetables do not appear to be a common reservoir of these bacteria, although in one study *H. alvei* was recovered from some local and imported vegetable samples [8,9]. There are only a few anecdotal reports describing the isolation of *H. alvei* from water [8]. *Serratia* sp., however, is the dominant genus of bacteria in fish and the main cause of spoilage of raw products and those stored in a modified atmosphere [6,10] Many species are associated with food spoilage, and some of them (mainly *S. marcescens* and *S. liquefaciens*) have been described as opportunistic human pathogens [11]. The main diseases caused by Serratia include urinary tract and central nervous system infection, bacteremia, pneumonia and, less commonly, eye infections. In the case of *H. alvei*, infections, affecting immunocompromised patients with underlying conditions, manifest mostly as diarrhea or bacteremia. Infections occur mainly through the dirty hands of healthcare professionals, but can also cause pneumonia through ventilation, catheter-related urinary tract infections, infections caused by parenteral nutrition, but also through the digestive system [12,13,14]. As with bacteria of all genera, there is an increasing prevalence of antibiotic-resistant strains among *Enterobacterales*, especially the β-lactam type, which is dominated by the mobilization of single, continuously expressed genes that code for efficient drug-modifying enzymes. The widespread occurrence of resistance strains is mostly reported in clinical settings; however, ESBL/AmpC-producing bacteria are increasingly being reported from livestock, companion animals, and environmental sources [15,16]. Wide dissemination of AMR genes in Enterobacteriales is mainly achieved by plasmids belonging to various incompatibility groups (Inc) such as F, A/C, L/M, I1, HI2, and N1 [17]. Indeed, the most striking feature of the environmental microbiome is its enormous diversity, providing numerous genes that could potentially be acquired and used by pathogens to counteract the action of antibiotics. Thereby the control of antimicrobial resistance in the environment is so important [18], especially in fish, as data on the amount of antibiotics used in aquaculture are scarce as few countries, generally those of Northern Europe, North, America and Japan, monitor the amount of antibiotics used [19]. Therefore, due to the real threat to health and life and the important issue of antimicrobial resistance among opportunistic pathogens, the aim of the study was to determine the antibiotic resistance profile of non-pathogenic *Klebsielleae* tribe strains.

## 2. Materials and Methods

### 2.1. Strains

Forty-seven strains belonging to the *Klebsiellae* tribe, thirteen strains of *H. alvei* and thirty-four *Serratia* sp. strains isolated from raw fish (*Salmo salar* and *Oncorhynchus mykiss*) and shrimp (*Penaeus monodon*) were used in this study. All strains were identified using MALDI-TOF as described in previous work [20]. Briefly, strains were incubated on Blood Agar (Biomaxima, Lublin, Poland) for 24 h at 37 °C, then single colonies wee spread evenly on a Disposable Target Slide (bioMérieux, Marcy-l’Étoile, France) then 1 µL of matrix (MS-CHCA, bioMérieux, Marcy-l’Étoile, France) was added and samples were left until completely dry. Prepared slides were transfer to VITEK-MS (bioMérieux, Marcy-l’Étoile, France) and analyzed using Saramis software (bioMérieux, Marcy-l’Étoile, France). All of them were obtained from the microbiological collection of the Department of Industrial and Food Microbiology University of Warmia and Mazury in Olsztyn and are listed in Table 1. Prior to analysis the isolates were stored in a Microbank at −80 °C (Biocorp, Lublin, Poland). After storage, and immediately before the desired use strains were grown from frozen stocks in 5 mL of Brain Heart Infusion broth (Merck, Darmstadt, Germany) overnight at 37 °C.

### 2.2. Antibiotic Resistance Profile

Susceptibility to antibiotics based on an automated minimal inhibitory concentration (MIC) was determined on a VITEK^®^ 2 Compact System instrument (bioMérieux, Marcy-l’Étoile, France) using VITEK^®^ AST N330 susceptibility card for Gram negative organisms, according to the manufacturer’s instructions. Briefly, bacterial inoculum was prepared from pure overnight culture grown on Trypticase Soy Agar (Merck, Darmstadt, Germany) by suspending single colonies in sterile saline solution to an optical density of 0.5–0.63 McFarland units as measured by DensiCheck Plus (bioMérieux, Marcy-l’Étoile, France). Tube with suspension was then placed in the appropriate slot of the VITEK 2 Compact system and analysis was performed by the instrument. The card contained 17 antibiotics: Amikacin (AK), Cefotaxime (CTX), Cefoxitin (FOX), Cefalexin (CN), Meropenem (MEM), Cefuroxime (CXM), Gentamicin (GEN), Ciprofloxacin (CIP), Norfloxacin (NOR), Ceftazidime (CAZ), Cefepime (FEP), Ertapenem (ETP), Nitrofurantoin (FT), Piperacillin/Tazobactam (TZP) and Trimpethoprim/sulphametaxazole (SXT). Interpretation of MIC values obtained on VITEK 2 was made according to the CLSI M100 guidelines [21]. Quality control was performed according to the manufacturer’s instruction for VITEK testing with reference strain *Escherichia coli* ATCC 25922.

### 2.3. Detection of ESBL-Positive Isolates

The extended-spectrum beta-lactamase (ESBL) producing strains were confirmed using MIC test strips (Liofilchem, Roseto degli Abbruzzi, Italy) containing Cefotaxime and Cefotaxime + Clavulanic acid (CTX/CTL) Ceftazidime/Ceftazidime + Clavulanic acid (CAZ/CAL) Cefepime/Cefepime + Clavulanic acid (FEP/FEL). The procedure was performed according to the manufacturer’s instructions, shortly cultures after 24 h incubation were used to make a 0.5 McFarland standard turbidity, then with a sterile swab culture were streak over the Mueller Hinton agar surface (Merck, Darmstadt, Germany). A few minutes after liquid absorbance, MIC Test Strip ESBLs were applied. The agar plates were incubated at 35 ± 2 °C for 16–20 h in ambient atmosphere. After incubation results were interpreted according to manufacturer guidelines. *K. pneumoniae* ATCC^®^ 700,603 strain was a positive control.

### 2.4. Detection of AmpC-Positive Isolates

Test strips containing: Cefotetan (CTT)/Cefotetan + Cloxacillin (CXT), Ertapenem (ETP)/Ertapenem + Cloxacillin (ECX) AND Ertapenem (ETP)/Ertapenem + Phenylboronic acid (EBO) (Liofilchem, Roseto degli Abbruzzi, Italy) were used to detect AmpC-positive strains, the same as for ESBL-positive strain detection. *K. pneumoniae* ATCC^®^ BAA-1144 was used as a positive control.

### 2.5. Detection of KPC-Positive Isolates

Test trips containing Ertapenem (ETP)/Ertapenem + Phenylboronic acid (EBO) or Meropenem (MRP)/Meropenem + Phenylboronic acid (MBO) (Liofilchem, Roseto degli Abbruzzi, Italy) were used to detect *Klebsiella pneumoniae* carbapenemases (KPC) positive strains; the procedure was the same as for ESBL-positive strains detection. *Stenotrophomonas maltophilia* ATCC^®^ 13636 was used as a positive control.

### 2.6. Detection of MBL-Positive Isolates

For the detection of Metallo-β-Lactamase (MBL) producing strains, test strips containing: Imipenem (IMI)/Imipenem + EDTA (IMD) or Meropenem (MRP)/Meropenem + EDTA (MRD) (Liofilchem, Roseto degli Abbruzzi, Italy) were used; the procedure was the same as for the detection of ESBL-positive strains. *K. pneumoniae* ATCC*^®^* BAA-1705 was used as a positive control.

## 3. Results

As a result of the conducted research, it was observed that all the tested *H. alvei* strains showed sensitivity to cefoxitin, cefepime, ertapenem, meropenem, amikacin, gentamicin, ciprofloxacin, norfloxacin and trimethoprim-sulfamethoxazole. The antibiotic to which the most strains (84.6%) showed resistance was cephalexin. A slight percentage of the strains were resistant to piperacillin/tazobactam (23.1%), cefuroxime (30.8%) and cefoxitin (7.7%) and, in four strains, intermediate resistance to cefuroxime, ceftazidime and nitrofurantoin was observed. Among all, just one strain was AmpC positive (Figure 1).

Among the tested strains of *Serratia* sp., 18 (58.0%) were defined as multi-drug resistant, resistant to minimum one agent in three or more chemical classes of antibiotic. The antibiotic to which the strains most frequently showed resistance was nitrofurantoin (85.3%). Antibiotics to which none of the tested strains showed resistance included piperacillin, cefoxatim, cefepime, ertapenem, meropenem, gentamicin, ciprofloxacin and norfloxacin. The largest number of *S. fonticola* strains were resistant to cefalexin (63.6%) and cefuroxime (72.7%), while *S. liquefaciens* were resistant to cefuroxime (88.2%) and nitrofurantoin (94.1%). The strains that were resistant, intermediate, and susceptible to individual antibiotics are shown in Figure 2.

## 4. Discussion

This is the first report describing antibiotic susceptibility among non-pathogenic strains belonging to the tribe *Klebsielleae.* In the conducted study it was observed that most strains of the *H. alvei* were cefalexin resistant, followed by piperacillin/tazobactam and cefuroxime. Researchers observed resistance to antibiotics such as cefazolin, amoxicillin/clavulanic acid, ampicillin/sulbactam, ciprofloxacin, aminoglycosides, ceftriaxone, cefotaxime, imipenem, ticarcillin, piperacoxillazole and co-trimazole in the clinical strains [7,22,23]. In a study conducted by Yin et al., DNA sequencing of 13 strains of *H. alvei* was performed, and the results showed that the strains contained several resistance genes coding nonspecific antimicrobial resistance, including several genes encoding efflux pumps. The susceptibility profile of *Hafnia* sp. in this study highlighted prevalent resistance to amoxicillin-clavulanic acid and colistin, intermediate resistance to piperacillin-tazobactam, ceftriaxone, ceftazidime and ertapenem [24], which partially covers the phenotypic results in the current study. Moreover, none of the tested *Hafnia* strains were ESBL, MBL, or KPC-positive, and only one of the strains was AmpC-positive. In a study of Enterobacterales strains from retail products, it was shown that all (*n* = 2) strains of *H. alvei* were AmpC-producers, while none were ESBL-producers [25]. However, in a study of clinical strains, up to 50% were the ESBL-producing strains [26]. In the case of *Serratia* sp. strains, the vast number of strains were resistant or intermediate to cefalexin, cefuroxime and nitrofurantoin. On the other hand, a study of fish intestines from Turkey showed that most strains of *Serratia* sp. were resistant to ampicillin and piperacillin [11]. When examining the clinically significant species of the genus *Serratia* (*S. marcescens*), most strains were resistant to ceftriaxone, ceftazidime and piperacillin/tazobactam [27]. In addition, it was observed that 3.7% of *S. marcescens* strains are ESBL-positive, and other studies have shown that six out of 23 *S. marcescens* strains were MPC-positive [28]. Due to the clinical significance of *S. marcescens*, among others, *Serratia* sp. DNA sequencing research has focused on these strains, and it has been shown that all strains contain genes responsible for resistance to aminoglycoside (AAC(6′)-Ic) and AmpC β-lactamase (CRP^b^) [29]. 

In studies of strains that cause infections in humans, it can be observed that ciprofloxacin, piperacillin/tazobactam or trimethoprim/sulfamethoxazole [25,26] were used to combat *H. alvei*, while in the case of patients infected with *Serratia* sp. cefepime, ceftriaxone, piperacillin-tazobactam, levofloxacin and clindamycin [25] were applied. In a study of clinical Enterobacterales strains, it was determined that piperacillin with tazobactam was the most commonly used [30]. The use of TZP in the UK has increased yearly, from less than 2.1% of all antibiotics prescribed in 2008–2009 to 3.6% in 2012–2013 [31,32]. Overuse of antibiotics is one of the main causes of the growing number of resistant strains [33]; however, in the current study, none of the *Serratia* spp. strains were resistant, while only 23% of *H. alvei* strains showed resistance to TZP. The reason may be that the mutations appear and become more permanent in the patients or animals treated with antibiotics due to stronger selective pressure on pathogens. In contrast, environment and food strains are generally less likely to develop resistance to antimicrobials; however, the processes are not known yet [34]. However, these niches make an important contribution to the evolution of mutation-based resistance to most of the pathogens. Regarding the absorption of new immunity factors, water, soil, food, and other environments with highly variable ecological niches provide an unmatched gene pool with a diversity that far surpasses the microbiota of humans and domestic animals. On the other hand, from the epidemiological point of view, the high concentration of microorganisms in the guts promotes the transfer of resistance on mobile elements due to transformation, transformation, conjugation and gene transfer agents. Research shows that the genomes of gut-adapted bacteria are more comparable in terms of gene content over a given evolutionary distance than the non-enteric genomes. Thus, shared functional needs or increased horizontal transfer result in similarities between genes within the gut. It has been theorized that the mammalian gut is “a melting pot of genetic exchange, which causes a large range of horizontal gene transfer to occur” [35,36,37].

## 5. Conclusions

Antibiotic resistance is a natural adaptive process of bacteria that predates modern human impact on the environment and may have been accelerated by the ubiquitous presence of antimicrobial compounds in medicine, agriculture and the environment. Although the occurrence of resistant bacteria is more frequent in clinical strains, due to the wide variety and ease of transmission of resistance genes, environmental strains must be continuously monitored. Additionally, attention should be focused not only on pathogens but also on opportunistic pathogens. The high use of antibiotics in animal husbandry and the lack of control in some countries, especially in the fish industry, should be a key indicator of the prevalence of antibiotic-resistant strains. It is reassuring that strains of the genus *Serratia* and *Hafnia alvei*, frequently isolated from fish, are often sensitive to clinically important antibiotics and do not have resistance mechanisms typical for the tribe *Klebsielleae*.

## Figures and Tables

**Figure 1 foods-11-02270-f001:**
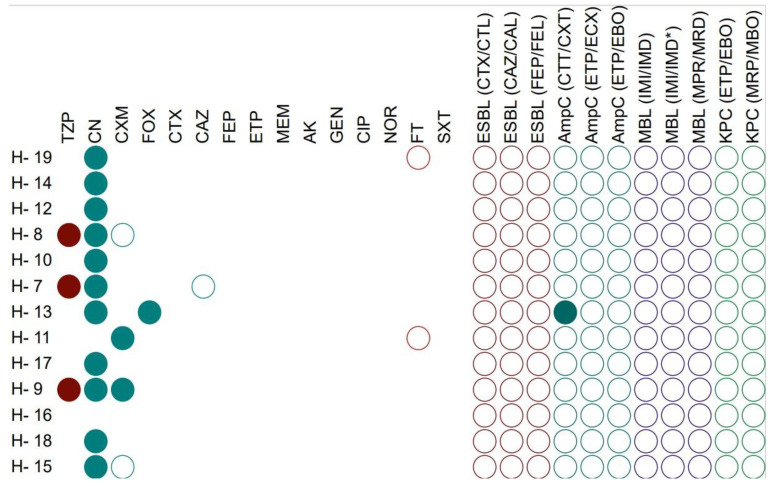
Antibiotic resistance profile of *H. alvei* strains; amikacin (AK), cefotaxime (CTX), cefoxitin (FOX), cephalexin (CN), meropenem (MEM), cefuroxime (CXM), gentamicin (GEN), ciprofloxacin (CIP), norfloxacin (NOR), ceftazidime (CAZ), cefepime (FEP), ertapenem (ETP), nitrofurantoin (FT), piperacillin/tazobactam (TZP) and trimethoprim/sulfamethoxazole (SXT), cefotaxime and cefotaxime + clavulanic acid (CTX/CTL), ceftazidime/ceftazidime + clavulanic acid (CAZ/CAL), cefepime/cefepime + clavulanic acid (FEP/FEL), cefotetan (CTT)/cefotetan + cloxacillin (CXT), ertapenem (ETP)/ertapenem + cloxacillin (ECX), ertapenem (ETP)/ertapenem + phenylboronic acid (EBO), ertapenem (ETP)/ertapenem + phenylboronic acid (EBO), meropenem (MRP)/meropenem + phenylboronic acid (MBO, imipenem (IMI)/imipenem + EDTA (IMD), meropenem (MRP)/meropenem + EDTA (MRD). For antibiotics: filled shape-resistance, shape outline-intermediate, shape completely omitted-sensitive, for mechanisms: filled shape-positive, shape outline-negative. Strains antibiogram visualized using iTOL.

**Figure 2 foods-11-02270-f002:**
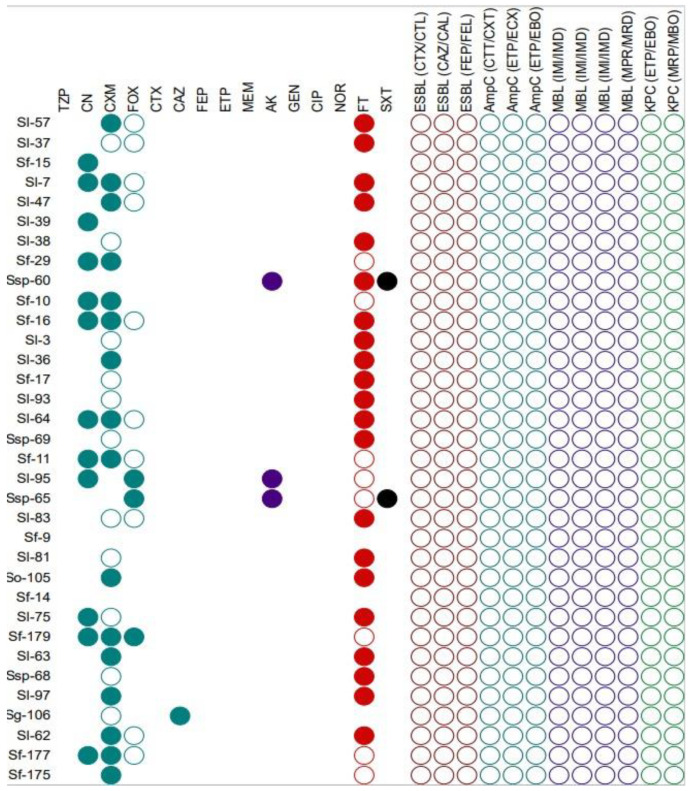
Antibiotic resistance profile of *Serratia* sp. strains; amikacin (AK), cefotaxime (CTX), cefoxitin (FOX), cephalexin (CN), meropenem (MEM), cefuroxime (CXM), gentamicin (GEN), ciprofloxacin (CIP), norfloxacin (NOR), ceftazidime (CAZ), cefepime (FEP), ertapenem (ETP), nitrofurantoin (FT), piperacillin/tazobactam (TZP) and trimethoprim/sulfamethoxazole (SXT), cefotaxime and cefotaxime + clavulanic acid (CTX/CTL), ceftazidime/ceftazidime + clavulanic acid (CAZ/CAL), cefepime/cefepime + clavulanic acid (FEP/FEL), cefotetan (CTT)/cefotetan + cloxacillin (CXT), ertapenem (ETP)/ertapenem + cloxacillin (ECX), ertapenem (ETP)/ertapenem + phenylboronic acid (EBO), ertapenem (ETP)/ertapenem + phenylboronic acid (EBO), meropenem (MRP)/meropenem + phenylboronic acid (MBO, imipenem (IMI)/imipenem + EDTA (IMD), meropenem (MRP)/meropenem + EDTA (MRD). For antibiotics: filled shape-resistance, shape outline-intermediate, shape completely omitted-sensitive, for mechanisms: filled shape-positive, shape outline-negative. Strains antibiogram visualized using iTOL.

**Table 1 foods-11-02270-t001:** Strains and their source of isolation.

Species	Source	No. of Isolates	Level of Identity [%]
***Hafnia* sp.**		**13**	
*H. alvei*	salmon (*Salmo salar*)	9	91.8–99.9
	trout (*Oncorhynchus mykiss*)	4	99.1–99.9
***Serratia* sp.**		**34**	
*S. fonticola*	salmon (*Salmo salar*)	2	99.9
	trout (*Oncorhynchus mykiss*)	5	90.1–99.9
	prawn (*Penaeus monodon*)	5	97.1–99.9
*S. grimensi*	salmon (*Salmo salar*)	1	99.9
*S.liquefaciens*	salmon (*Salmo salar*)	9	95.6–99.9
	trout (*Oncorhynchus mykiss*)	4	92.1–99.9
	prawn (*Penaeus monodon*)	3	99.9
*S. quinivorans*	salmon (*Salmo salar*)	1	99.9
Other	trout (*Oncorhynchus mykiss*)	4	99.9

## Data Availability

The dataset used during this study is the available form given author upon reasonable request.

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
