# Peer review of "Short Communication: Low Prevalence of Clinically Important Antibiotic-Resistant Strains among Non-Pathogenic Genera of the Tribe Klebsielleae"

_foods, 2022, doi:10.3390/foods11152270_

Round 1
Reviewer 1 Report
Although most of the points raised have been addressed, the manuscript still need minor revisions:
Line 51. Enterobacteriales.
Line 55. Insert full stop after “antibiotics”, start the next sentence with “Thereby the control of antimicrobioal resistance in environmental is so important .......[17].
Line 64. 32 Serratia isolates in the manuscript vs. 34 in the table 1.
Line 188. Please replace “intermidiate” with “intermediate resistance”
Line 210. Replace “resistant” with “resistance”, Line 211 “know” to “known”.
Line 212. Most of the pathogens.
Line 223. Replace “strains” with “pathogens”.
Line 226. add “clinically” before “important”
Line 227. Insert “resistance” prior to “mechanisms”
Author Response
All changes proposed by Reviewers in manuscript are clearly highlighted, using the "Track Changes" function in Microsoft Word
Although most of the points raised have been addressed, the manuscript still need minor revisions:
Line 51. Enterobacteriales.
Thank you, misspelling has been corrected
Line 55. Insert full stop after “antibiotics”, start the next sentence with “Thereby the control of antimicrobial resistance in environmental is so important .......[17].
Thank you for suggestion, it has been corrected
Line 64. 32 Serratia isolates in the manuscript vs. 34 in the table 1.
Numbers has been unified
Line 188. Please replace “intermidiate” with “intermediate resistance”
It has been corrected
Line 210. Replace “resistant” with “resistance”, Line 211 “know” to “known”.
Thank you, misspelling has been corrected
Line 212. Most of the pathogens.
Thank you we followed your suggestion
Line 223. Replace “strains” with “pathogens”.
Thank you we followed your suggestion
Line 226. add “clinically” before “important”
we followed your suggestion
Line 227. Insert “resistance” prior to “mechanisms”
Thank you it has been corrected
Reviewer 2 Report
The authors presented a study of antibiotic resistant bacteria that usually are non-pathogenic or are rare causing diseases, however the importance of this kind of studies should be better highligted. The use of systematic nomenclature, in my opinion, should be harmonized. They use order in several times but in one ocasion use family, I would prefer this last one. They spell Enterobacteriales and Enterobacterales; Klebsielleae and Klebsiellae.
Please verify some spelling errors in the abstract and verify if it is correct to say Enterobacterales-AmpC or if it should be AmpC b-lactames-producing Enterobacterales.
In the introduction please verify the sentence on line 38-40 concerning Serratia as the main organism causing fish spoilage. If possible add more references that also confirm this.
Lines 40 - 44 mention Serratia as opportunistic human pathogen and after describe some diseases, and one is diarrhea. This last could be important to relate the work with the ingestion of foods carrying these microrganism. It might be explored.
Materials and methods
Line 63 mention 32 strains of Serratia instead of 34.
Line 94 missing reference or wrong reference?
Line 118 please write the name of the ESBL positive strain S. maltophilia.
Results
In figure 1 and 2 it is mentioned Phylogenetic tree, what do you mean?
Line 149 please define multi-drug resistance
Discussion
I think it can be improved and consider the interacion with intestinal human microbiome .
Author Response
All changes proposed by Reviewers in manuscript are clearly highlighted, using the "Track Changes" function in Microsoft Word
The authors presented a study of antibiotic resistant bacteria that usually are non-pathogenic or are rare causing diseases, however the importance of this kind of studies should be better highligted. The use of systematic nomenclature, in my opinion, should be harmonized. They use order in several times but in one ocasion use family, I would prefer this last one. They spell Enterobacteriales and Enterobacterales; Klebsielleae and Klebsiellae.
Nomenclature has been unified, I would prefer to avoid using Families names because Klebsiella, Enterobacter, Hafnia and Serratia belong to 3 different families.
Please verify some spelling errors in the abstract and verify if it is correct to say Enterobacterales-AmpC or if it should be AmpC b-lactames-producing Enterobacterales.
Thank you, I believe it is more informative, so we followed your suggestion
In the introduction please verify the sentence on line 38-40 concerning Serratia as the main organism causing fish spoilage. If possible add more references that also confirm this.
We have cited a strong conclusion from an article, however citation has been added.
Lines 40 - 44 mention Serratia as opportunistic human pathogen and after describe some diseases, and one is diarrhea. This last could be important to relate the work with the ingestion of foods carrying these microrganism. It might be explored.
Thank you, transmition of bacteria has been briefly described
Materials and methods
Line 63 mention 32 strains of Serratia instead of 34.
Thank you, it has been corrected
Line 94 missing reference or wrong reference?
Thank you, reference has been changed
Line 118 please write the name of the ESBL positive strain S. maltophilia.
Genus’s name has been added
Results
In figure 1 and 2 it is mentioned Phylogenetic tree, what do you mean?
I apologize for an oversight; unnecessary information has been removed
Line 149 please define multi-drug resistance
Thank you, definition has been added
Discussion
I think it can be improved and consider the interacion with intestinal human microbiome .
Thank you for your suggestion, we added information about horizontal gene transfer
This manuscript is a resubmission of an earlier submission. The following is a list of the peer review reports and author responses from that submission.
Round 1
Reviewer 1 Report
Zakrzewski et al. describe the investigation of prevalence of clinically important antibiotic-resistant strains from Klebsiellae tribe isolated from fish and shrimps. In general, the manuscript is well written and easy to comprehend, and the study design is appropriate. However, I have several comments to be addressed.
Major comments:
The authors built a phylogenetic tree for their isolates, but have not described the data used for the tree building. As far as I understood, the isolates were not subjected to whole genome sequencing. So please describe what the trees are built on – some gene sequences or what?
Minor comments:
- please fix the name of the tribe in the title. In addition, I think that ‘Short communication’ should not be included in the title
- line 15 – should be “a vast”, not “an vast”
- line 16 – please add dash or the like between “strains” and “to cefuroxime"
- line 17 – should be “one had”, not “one has” (wrong sequence of tenses)
- Chapter 2.1 – were the strains isolated from fresh fish and shrimps or from frozen products? Please describe.
- table 1 – please specify what ID means (genomic similarity or some other identity?)
Line 61-62 and 66-67 – ‘bioMerieux’ is repeated several times
Line 77 – please define ESBL at first occurrence of the abbreviation
Line 94 – please define KPC here
Line 98 – please define MBL here
Line 103 – “All data phylogenetic tree” – please rephrase, e.g., “Phylogenetic tree for all data” or “all isolates”
Line 164 – pleas italicize H. alvei
Line 207 – “it” is redundant in “it should”
Author Response
Major comments:
The authors built a phylogenetic tree for their isolates, but have not described the data used for the tree building. As far as I understood, the isolates were not subjected to whole genome sequencing. So please describe what the trees are built on – some gene sequences or what?
In methods it has been precised that tree was build based on spectra similarities
Minor comments:
- please fix the name of the tribe in the title. In addition, I think that ‘Short communication’ should not be included in the title
- line 15 – should be “a vast”, not “an vast”
Thank you, it has been corrected
- line 16 – please add dash or the like between “strains” and “to cefuroxime"
Thank you, it has been corrected
- line 17 – should be “one had”, not “one has” (wrong sequence of tenses)
Thank you, it has been corrected
- Chapter 2.1 – were the strains isolated from fresh fish and shrimps or from frozen products? Please describe.
Information has been precised
- table 1 – please specify what ID means (genomic similarity or some other identity?)
Thank you for comments, table has been improved
Line 61-62 and 66-67 – ‘bioMerieux’ is repeated several times
Thank you, it has been corrected
Line 77 – please define ESBL at first occurrence of the abbreviation
Thank you full names have been added
Line 94 – please define KPC here
Thank you full names have been added
Line 98 – please define MBL here
Thank you full names have been added
Line 103 – “All data phylogenetic tree” – please rephrase, e.g., “Phylogenetic tree for all data” or “all isolates”
Thank you, it has been corrected
Line 164 – pleas italicize H. alvei
Thank you, it has been corrected
Line 207 – “it” is redundant in “it should”
Thank you, it has been corrected

Reviewer 2 Report
It was my pleasure to review the manuscript by Zakrzewski et al., on antimicrobial resistance patterns in Enterobacteriales isolated from fish. This is an interesting study, however further clarifications are needed.
Title
How was the clinical importance of the strains assessed? All the strains were isolated from fish so they could be of veterinary importance or the origin of stains, ie. Fish-borne isolates could be emphasized.
Non-pathogenic – how was the pathogenicity determined? If the pathogenicity was not detected, please omit it.
Line 9. Delete “Both genera”, start sentence with Hafnia.
Line 14. Provide full names for abbreviations at the first mention.
Provide numbers on the occurrence of resistant and susceptible strains.
Line 18-20. Control of antibiotic-resistant strains. How could this be controlled in practice? Replace with ie. “could be an issue or concern….
Introduction.
The authors describe the properties of Hafnia and Serratia, however do not provide details on the origins and properties of enzyme-producing bacteria, their importance in aquaculture.
Line 26. Number of reports?
Line 27. Insert the appropriate reference after ‘genera”, delete “both.
Line 30. Products of animal origin? Delete “meat as”.
Line 33. Please provide the appropriate reference after “samples”.
Line 33. Why anecdotal? Report or reports? Only one reference is provided.
Line 35. Dominant spoilage bacteria?
Line 40. Diarrhea is a clinical manifestation/ symptom
Line 40. Delete “As….genera”.
The aim of the study needs to be more focused on the research question but not on the description of previously conducted research.
Materials.
Table 1. no. No. of isolates?
What is the level of ID? Please provide the details on the methods applied for microbial identification.
Explanation of indexes in the last column is needed.
Add common names for fish species.
Please provide full names for all abbreviations at the first mention.
Were the strips for detection of Amp, KPC, MBL commercially produced? Please provide any details?
Results.
Line 112. Mechanism of resistance was not studied, only the occurrence of AmpC resistant isolates was detected.
Figure 1. I recommend not to leave empty spaces.
“Mechanisms”. They are not mechanisms but different types of enzymes produced by resistant bacteria.
Dendrogram does not contain the information about the similarity of the isolates , ie. horizontal line with the similarity level.
Line 134. What are variable resistant strains?
Figure 2. See my comment for Figure 1.
Discussion.
Line 151. Delete “it … that”
Line 152. Replace “are” with “were”.
Line 153-154. Are those the findings of the present study?
Line 156. Delete “was… study”, replace “that… contained” with “the presence of”.
Line 158-161. Not clear, rewrite.
Line 161. What is “partial resistance”?
Line 167. Resistant and moderately resistant? How do they differ from the partial resistance? Please define what you mean with those definitions?
Line 175. Could the genes responsible for resistance to aminoglycoside be added?
Line 177-180. Do the authors describe the pattern of resistance in clinical strains?
Line 184. Overuse of antibiotics in medicine?
In my opinion, the discussion is unfocused. The author describes the results from clinical studies while the research covers the resistance of Enterobacteriales from the aquatic environment. Lines 193-198 perfectly fit for the introduction to highlight the importance of the present study.
Conclusions.
Lines 209-210. Only this sentence contains the main findings relevant to the present study while the mechanisms of resistance were not covered.
Author Response
It was my pleasure to review the manuscript by Zakrzewski et al., on antimicrobial resistance patterns in Enterobacteriales isolated from fish. This is an interesting study, however further clarifications are needed.
Title
How was the clinical importance of the strains assessed? All the strains were isolated from fish so they could be of veterinary importance or the origin of stains, ie. Fish-borne isolates could be emphasized.
By clinically important strains authors meant species define as pathogens. Additionally, the research was conducted in the Department of Industrial and Food Microbiology and we perceive fish as food and potential source of food-born diseases
Non-pathogenic – how was the pathogenicity determined? If the pathogenicity was not detected, please omit it.
By non-pathogenic authors meant all species do not consider as pathogens for humans
Line 9. Delete “Both genera”, start sentence with Hafnia.
Thank you for your comment, words were deleted
Line 14. Provide full names for abbreviations at the first mention.
Full names for abbreviations have been added
Provide numbers on the occurrence of resistant and susceptible strains.
Thank you, I have provided it
Line 18-20. Control of antibiotic-resistant strains. How could this be controlled in practice? Replace with ie. “could be an issue or concern….
Thank you for your comment
Introduction.
The authors describe the properties of Hafnia and Serratia, however do not provide details on the origins and properties of enzyme-producing bacteria, their importance in aquaculture.
Thank you for your comment, the reference to aquaculture has been added
Line 26. Number of reports?
Unfortunately, we are not able to determine the exact number of reports, we wrote this on the basis of articles such as:
Gupta Varsha, Sharma Shiwani *, Pal Kritika , Goyal Poonam , Agarwal Deepak and Chander Jagdish , Serratia, No Longer an Uncommon Opportunistic Pathogen – Case Series & Review of Literature, Infectious Disorders - Drug Targets 2021; 21(7) : e300821191666 . https://dx.doi.org/10.2174/1871526521666210222125215
Serratia, No Longer an Uncommon Opportunistic Pathogen – Case Series & Review of Literature
Line 27. Insert the appropriate reference after ‘genera”, delete “both.
Thank you, the text has been corrected
Line 30. Products of animal origin? Delete “meat as”.
Thank you, the text has been corrected
Line 33. Please provide the appropriate reference after “samples”.
Reference has been added
Line 33. Why anecdotal? Report or reports? Only one reference is provided.
Thank you for your comment, we decided to cite just revew article because some of reference there are outdated
Line 35. Dominant spoilage bacteria?
In cited work, identification based on DNA sequencing was performed and the results showed that it is indeed dominant spoilage bacteria
Line 40. Diarrhea is a clinical manifestation/ symptom
Thank you, line has been corrected
Line 40. Delete “As….genera”.
Thank you, the text has been corrected
The aim of the study needs to be more focused on the research question but not on the description of previously conducted research.
Thank you for the comment, additional information has been omitted
Materials.
Table 1. no. No. of isolates?
What is the level of ID? Please provide the details on the methods applied for microbial identification.
Explanation of indexes in the last column is needed.
Thank you for comments, table has been improved
Add common names for fish species.
Thank you, names have been added
Please provide full names for all abbreviations at the first mention.
Thank you, full names have been added
Were the strips for detection of Amp, KPC, MBL commercially produced? Please provide any details?
Information has been added
Results.
Line 112. Mechanism of resistance was not studied, only the occurrence of AmpC resistant isolates was detected.
thank you, the text has been corrected
Figure 1. I recommend not to leave empty spaces.
Empty line has been removed
“Mechanisms”. They are not mechanisms but different types of enzymes produced by resistant bacteria.
The ability to produce enzymes is referred to as the resistance mechanism and is referred to in the literature as such
Thomson K. S. (2010). Extended-spectrum-beta-lactamase, AmpC, and carbapenemase issues. Journal of clinical microbiology, 48(4), 1019–1025. https://doi.org/10.1128/JCM.00219-10
Dendrogram does not contain the information about the similarity of the isolates , ie. horizontal line with the similarity level.
Graphical representation only, iTOL does not provide a horizontal line, only the length of branches that are not readable
Line 134. What are variable resistant strains?
Sorry, unfortunate use of word, in whole document it has been changed to intermediate
Figure 2. See my comment for Figure 1.
Thank you for comment
Discussion.
Line 151. Delete “it … that”
Thank you, it has been corrected
Line 152. Replace “are” with “were”.
Thank you, it has been corrected
Line 153-154. Are those the findings of the present study?
Yes, they are
Line 156. Delete “was… study”, replace “that… contained” with “the presence of”.
Thank you, it has been corrected
Line 158-161. Not clear, rewrite.
Thank you, it has been corrected
In a study conducted by Yin et al., DNA sequencing of 13 strains of H. alvei was performed, and the results showed that the strains contained several resistance genes coding nonspecific antimicrobial resistance, including several genes encoding efflux pumps
Line 161. What is “partial resistance”?
Sorry, unfortunate use of word, in whole document it has been changed to intermediate
Line 167. Resistant and moderately resistant? How do they differ from the partial resistance? Please define what you mean with those definitions?
Thank you for comments, I agree that use of those words was unfortunate, it was changed according to CLSI recommendation to intermediate
Line 175. Could the genes responsible for resistance to aminoglycoside be added?
Information has been added
Line 177-180. Do the authors describe the pattern of resistance in clinical strains?
No, we were referred to used antibiotics for patients with infections caused by Serratia and Hafnia
Line 184. Overuse of antibiotics in medicine?
Yes, it is about putting pressure on doctors to prescribe antibiotics, although they are not necessary, and the lack of regulation in many countries and the ability to buy them over the counter, also online
In my opinion, the discussion is unfocused. The author describes the results from clinical studies while the research covers the resistance of Enterobacteriales from the aquatic environment. Lines 193-198 perfectly fit for the introduction to highlight the importance of the present study.
Thank you for your opinions, I agree that the discussion is focused on clinical issues, but it is related to the fact that the antibiotic resistance crisis is only talked about in the context of animals and that strains isolated from food are treated as raw material for human consumption.
Conclusions.
Lines 209-210. Only this sentence contains the main findings relevant to the present study while the mechanisms of resistance were not covered.

Round 2
Reviewer 1 Report
The authors did not provide a reply to the comment regarding phylogenetic tree building. The only information in the manuscript is "Isolate spectra were grouped together in a taxonomy tree using the Saramis"
There is still no information on what spectra were used. What algorithm was used for building the tree? The readers might not be familiar with the software used, especially since it seems to be proprietary, so more detailed description on the data and methods used for building the tree is essential for understanding and justifying the conclusions.
Author Response
Thank you for your comment
The authors did not provide a reply to the comment regarding phylogenetic tree building. The only information in the manuscript is "Isolate spectra were grouped together in a taxonomy tree using the Saramis"
There is still no information on what spectra were used. What algorithm was used for building the tree? The readers might not be familiar with the software used, especially since it seems to be proprietary, so more detailed description on the data and methods used for building the tree is essential for understanding and justifying the conclusions.
Thank you for your comment, however, I am not able to extend this chapter, I added the information that the spectra come from isolate identification, however in terms of algorithms, Saramis software is not a mathematical software in which the user can operate with great freedom to create trees, this software is included to Vitek MS and allows you to compare strains and generate trees, but the producer does not provide any algorithms.

Reviewer 2 Report
Dear authors,
thank you for addressing my comments. Some clarifications and improvements are still needed.
Abstract
Line 12. Genera?
Line 17-18. Please provide a percentage for the resistant strains.
Line 18-19. The authors had not studied the resistance mechanism but evaluated the pattern of resistance.
Line 20. High resistance rates?
Line 21. The concluding sentence is not finished.
Introduction.
Line 28. “reports”, please provide more than one refence to support this idea.
Line 34-35. The authors report “reports” so please add more than one references to support your idea. Reference no.6 is a respectful clinical journal and I do not see any reason to claim this finding as anectodical.
Line 41. H. alvei is not a regular pathogen but the infection may occur in immunocompromised patient with underlying conditions. This needs to be specified.
Line 48. Enterobacteriales.
Line 53. What do you mean with “few countries”? Please specify.
Materials and methods.
Line 61. Given numbers do not correspond to those in the Table 1. Please check.
Line 94. Cultures after 24 h of incubation?
Line 96. Change “hours”to “h”.
2.4., 2.5., 2.6., start with “Test strips”.
Results.
Line 124. Provide percentage for cephalexin resistance.
Round all reported percentages to one decimal.
Please use the same style for reporting the results: how many were resistant/ total of the isolates, %.
Line 171, 180, 191. Please italicize Hafnia, S. marcescens
Line 186. Intermediate resistance?
Discussion
Line 207. Microorganisms develop antimicrobial resistance but not the environment or foods themselves. Also the appropriate reference is needed.
Nice figures but not very informative because an indication of the distance levels between the isolates is missing. Could the authors consider to add manually the missing information?
Author Response
Dear reviewer,
Thank you for your comments
Abstract
Line 12. Genera?
Plural form of genus
Line 17-18. Please provide a percentage for the resistant strains.
I’ve changed numbers for percentage
Line 18-19. The authors had not studied the resistance mechanism but evaluated the pattern of resistance.
Thank you, I’ve precised this information in abstract
Line 20. High resistance rates?
Thank you, it's a good point, I’ve changed it to high prevalent resistance
Line 21. The concluding sentence is not finished.
Thank you, it was corrected
Introduction.
Line 28. “reports”, please provide more than one refence to support this idea
The number of articles that mention about infections has been increased 1,2
Line 34-35. The authors report “reports” so please add more than one references to support your idea. Reference no.6 is a respectful clinical journal and I do not see any reason to claim this finding as anectodical.
Thank you, references has been added, and word has been removed
Line 41. H. alvei is not a regular pathogen but the infection may occur in immunocompromised patient with underlying conditions. This needs to be specified.
Information has been added
Line 48. Enterobacteriales.
The order’s name has been corrected
Line 53. What do you mean with “few countries”? Please specify.
Information has been added.
Materials and methods.
Line 61. Given numbers do not correspond to those in the Table 1. Please check.
Thank, it has been corrected.
Line 94. Cultures after 24 h of incubation?
Information has been precised
Line 96. Change “hours”to “h”.
It has been changed
2.4., 2.5., 2.6., start with “Test strips”.
It has been added
Results.
Line 124. Provide percentage for cephalexin resistance.
It has been provided
Round all reported percentages to one decimal.
It has been changed
Please use the same style for reporting the results: how many were resistant/ total of the isolates, %.
The results were unified
Line 171, 180, 191. Please italicize Hafnia, S. marcescens
It has been changed
Line 186. Intermediate resistance?
Yes it is correct with Clinical & Laboratory Standards Institute nomenclature
Discussion
Line 207. Microorganisms develop antimicrobial resistance but not the environment or foods themselves. Also the appropriate reference is needed.
It has been changed
Nice figures but not very informative because an indication of the distance levels between the isolates is missing. Could the authors consider to add manually the missing information?
Thank you, information has been added
